# Improved Algorithms for Clustering with Distance Oracles

**Anonymous**

## Abstract

Bateni *et al.* has recently introduced the *weak-strong distance oracle model* to study clustering problems in settings with limited distance information. Given query access to the strong-oracle and weak-oracle in the weak-strong oracle model, the authors design approximation algorithms for $k$-means and $k$-center clustering problems. In this work, we design algorithms with improved guarantees for $k$-means and $k$-center clustering problems in the weak-strong oracle model. The $k$-means++ algorithm is commonly used to solve $k$-means where complete distance information is available. One of the main contributions of this work is to show that $k$-means++ algorithm can be adapted to work in the weak-strong oracle model using only a small number of strong-oracle queries, which is the critical resource in this model. In particular, our $k$-means++ based algorithm gives a constant approximation for $k$-means and uses $O(k^2 \log^2 n)$ strong-oracle queries. This improves on the algorithm of Bateni *et al.* that uses $O(k^2 \log^4 n \log^2 \log n)$ strong-oracle queries for a constant factor approximation of $k$-means. For the $k$-center problem, we give a simple *ball-carving* based $6(1 + \varepsilon)$-approximation algorithm that uses $O(k^3 \log^2 n \log \frac{\log n}{\varepsilon})$ strong-oracle queries. This is an improvement over the $14(1 + \varepsilon)$-approximation algorithm of Bateni *et al.* that uses $O(k^2 \log^4 n \log^2 \frac{\log n}{\varepsilon})$ strong-oracle queries. To show the effectiveness of our algorithms, we perform empirical evaluations on real-world datasets and show that our algorithms significantly outperform the algorithms of Bateni *et al.*

## 1 Introduction

Clustering problems such as the $k$-means, $k$-median and $k$-center problems are often studied in *full information* settings where the embeddings of the $n$ points to be clustered are given as part of the input to the problem. The $k$-clustering problems are normally formulated as minimizing some objective function of the input. However, these problems often turn out to be NP-hard, and approximation algorithms are designed to solve these problems. However, for many machine learning applications, the assumption of complete knowledge of the dataset under consideration might be either infeasible or very expensive to meet. This motivates the study of clustering problems in the *partial information* setting where there is a tradeoff between the accuracy of the input data and the resulting cost of clustering. The more accurate you want the clustering solution to be, the cost of collecting accurate information about the dataset would become higher. Questions such as *what is the best computational guarantee (e.g., approximation factor) one can obtain within a budget*, or *what is the minimum cost for obtaining a target guarantee*, become meaningful. Such studies can also be interpreted as computing with noisy information and are not entirely new (e.g., Feige *et al.* [12]), and the problems considered in these settings range from sorting and searching to graph problems ([17],[18]).

Clustering problems have been studied in various partial information settings. In these works, one typically assumes access to an oracle that provides some meaningful information about the underlying clustering, and the goal is to obtain a good approximate clustering solution using only a small number of oracle queries. A number of works studied variants of this problem with different assumptions

about the oracle and the clustering objective. Liu and Mukherjee [19] give tight query complexity bounds for the cluster recovery problem with a membership-query oracle, that on a query with any two points, answers whether the points belong to the same cluster or not. When the goal is to obtain a good approximate clustering solution, a number of works [4, 20] study clustering problems assuming additional access to a noisy membership-query oracle, also known as the same-cluster query oracle. Gamlath *et al.* [14] study algorithms for $k$-means where noisy labels are provided for each input point using an adversarial or random perturbation of the labels.

In this work, we study clustering problems in the *weak-strong oracle model* introduced by Bateni *et al.* [6]. The weak-strong oracle model has access to an expensive strong-oracle giving exact distances between any two points and a cheaper weak-oracle that gives noisy answers for the distances. Given that meaningful vector embeddings of varied accuracy are now frequently available in most ML applications for various types of data, it makes sense to consider a noise model where the distances (indicating dissimilarity) between data items are available, and accurate information comes at a cost. This was the primary motivation behind the weak-strong oracle model given by [6]. Next, we describe more formally the clustering problems with access to distance oracles.

In the clustering with distance oracles problem, the algorithm is given as input the number $n$ of points in the dataset and the number $k$ of clusters. We assume that the input points $X$ belong to some metric space $(\mathcal{X}, d)$. However, we don't have direct access to their embeddings, and also we don't know the distances between any two of these points. We are given query access to distance oracles that, on a query with any two points in $\mathcal{X}$, respond with the distance between the two points. Each query has a cost associated with it, and we assume the strong-oracle queries to be much more expensive compared to the weak-oracle queries. We wish to design algorithms for clustering problems that make only a small number of queries to these distance oracles and return good approximate solutions. The first distance oracle we study is the strong-oracle $SO$ that always returns the exact distance $d(x, y)$ for any two query points $x, y \in \mathcal{X}$.

**Definition 1.1** (Strong-oracle model). *In this model, an algorithm is given access to a strong-oracle that for any $x, y \in \mathcal{X}$, returns $SO(x, y) = d(x, y)$, the exact distance between the two points.*

We first observe that $O(nk)$ strong-oracle queries suffice to run the $k$-means++ algorithm, and using known results, one gets a constant factor bi-criteria approximation for $k$-means [2] using $O(nk)$ strong-oracle queries. We also show that $O(nk)$ strong-oracle queries are enough for a 2-approximation for $k$-center problem [15]. However, as mentioned earlier, access to such strong-oracles might be limited and quite expensive in practice. This motivates us to design algorithms that access a much cheaper and possibly erroneous weak-oracle $\widetilde{WO}$. For some fixed $\delta \in (0, 1/2)$, given two points $x, y \in \mathcal{X}$, the weak-oracle $\widetilde{WO}$ returns the exact distance $d(x, y)$ between the two points independently with probability $(1 - \delta)$ and returns an arbitrary answer with the remaining probability.

Moreover, we also assume that the query answers of the weak-oracle $\widetilde{WO}$ are *persistent* in the sense that the weak-oracle always returns the same answer to a query, even if asked multiple times.

**Definition 1.2** (Weak-oracle model). *In this model, an algorithm has access to a weak-oracle $\widetilde{WO}$ that for any two points $x, y \in \mathcal{X}$, outputs $d(x, y)$ with probability $1 - \delta$, and with remaining probability $\delta$, outputs an arbitrary value.*

**Weak-strong oracle model** of Bateni *et al.* [6]: An algorithm in this model is allowed to query both the weak-oracle and the strong-oracle. We show that algorithms with good guarantees can be designed for clustering problems in this model that use only a small number of strong-oracle queries.

## 1.1 Clustering problems

Let $X$ be a set of $n$ points in a metric space $(\mathcal{X}, d)$. The $k$-means cost of $X$ with respect to any set $C \subset \mathcal{X}$ is given as $\Phi(X, C) = \sum_{x \in X} d^2(x, C)$, where $d(x, C) = \min_{c \in C} d(x, c)$. The $k$-means clustering problem is defined as follows: *Given a set $X$ of $n$ points in a metric space $(\mathcal{X}, d)$ and an integer $k$, the objective in the $k$-means clustering problem is to output a set $C$ of $k$ centers for which the $k$-means cost function $\Phi(X, C)$ is minimized.* The $k$-means clustering problem is very well studied theoretically and has numerous applications in practice. This problem is known to be NP-hard [9]. The $k$-means++ algorithm is often used to solve $k$-means in practice [5]. The main idea is to iteratively sample $k$ points using a distribution called $D^2$-distribution that is updated after every iteration and depends on the previously chosen centers. It is known that $k$-means++ gives

$O(\log k)$-approximation in expectation for $k$-means [5]. Moreover, it can also be adapted to obtain a constant factor bi-criteria approximation [2] simply by oversampling centers (i.e., sample more than $k$ centers). We shall refer to this as oversampling-$k$-means++. An $(\alpha, \beta)$-bi-criteria approximation uses at most $\alpha k$ centers and returns a solution of cost at most $\beta$ times the optimal $k$-means solution. The $(k, z)$-clustering problem is a generalization of the $k$-means problem. In the $(k, z)$-clustering problem, given a set $X$ of $n$ points from a metric space and an integer $k$, the objective is to return a set $C \subset X$ of $k$ centers for which $\sum_{x \in X} d(x, C)^z$ is minimized. For $z = 2$, this is the $k$-means problem and for $z = 1$, this is the $k$-median problem.

Another way to formulate data clustering is using the $k$-center problem, where the goal is to optimize the following cost function. The $k$-center cost of $X$ with respect to a set $C \subset \mathcal{X}$ is defined as $\phi(X, C) = \max_{x \in X} d(x, C)$, where $d(x, C) = \min_{c \in C} d(x, c)$. More specifically, the $k$-center problem is defined as follows: *Given a set $X$ of $n$ points in a metric space $(\mathcal{X}, d)$ and an integer $k$, the objective in the $k$-center problem is to output a set $C$ of $k$ centers for which the $k$-center cost function $\phi(X, C)$ is minimized.* In other words, the goal is to find $k$ points such that the radius of the balls centered at these points that cover $X$, is minimized. The $k$-center problem is known to be NP-hard and 2-approximation algorithms are known for the $k$-center problem [15].

## 1.2 Main results

**Strong-oracle model:** We note that strong-oracle model is precisely the well-known classical model. So, let us remind ourselves of what is known. We observe that $k$-means++ algorithm can be executed using $O(nk)$ strong-oracle queries and using known results, one can show that it gives an $O(\log k)$-approximation in expectation for the $k$-means problem [5]. Further, the oversampling version of $k$-means++ gives constant factor bi-criteria approximation for $k$-means using only $O(nk)$ strong-oracle queries [2]. We also note that a $\Omega(nk)$ strong-oracle query lower bound for any constant factor approximation of $k$-means follows from Mettu and Plaxton [21]. Even though we state results only for the $k$-means problem, it is possible to extend these ideas to obtain a $O(2^{2z})$ bi-criteria approximation in expectation for the $(k, z)$-clustering using $O(nk)$ strong-oracle queries [26]. For the $k$-center problem, we observe that $O(nk)$ strong-oracle queries suffice to run Gonzalez's farthest-point algorithm to obtain a 2-approximation [15]. These results are discussed in Appendix D.

**Weak-strong oracle model:** We design algorithms for $k$-means and $k$-center problems in this model that use only a small number of strong-oracle queries while giving good approximate solutions. Our results improve upon the work of Bateni *et al.* [6]. We next state our main results for $k$-means and $k$-centers problems, and describe them in detail in Section 2 and Section 3.

**Theorem 1.1** (Upper bound for $k$-means). *There exists a constant factor approximation algorithm for $k$-means problem that makes $O(k^2 \log^2 n / \epsilon^6)$ strong-oracle queries and $O(nk \log n / \epsilon^3)$ weak-oracle queries in the weak-strong oracle model and succeeds with some constant probability.*

**Theorem 1.2** (Upper bound for $k$-center). *There exists a $6(1 + \epsilon)$-approximation algorithm for the $k$-center problem that makes $O\left(k^3 \log^2 n \log \frac{\log n}{\varepsilon}\right)$ strong-oracle and $O\left(nk \log n \log \frac{\log n}{\varepsilon}\right)$ weak-oracle queries in the weak-strong oracle model and succeeds with probability at least $(1 - 1/n^4)^2$.*

**Remark 1.1.** *We obtain the above results when the failure probability of the weak-oracle is $\delta = 1/3$. We note that our algorithms can be extended to work with any $0 < \delta < 1/2$. The dependency of $\delta$ on the sample complexity would become $1/(1/2 - \delta)^2$ [6].*

**Weak-oracle model:** We show a lower bound result in the weak-oracle model proving that any constant factor approximate clustering solution for $k$-means in this model requires to make $\Omega(nk/(1 - 2\delta)^2)$ weak-oracle queries. We prove the following result in Appendix F.

**Theorem 1.3.** *Let $\delta \in (0, 1/2)$. Any randomized algorithm giving a $O(1)$-approximation for $k$-means in weak-oracle model with probability at least $3/4$ requires expected $\Omega(nk/(1 - 2\delta)^2)$ queries.*

**Our contributions and comparisons with known works** In this work, we provide improved algorithms for clustering problems in the weak-strong oracle model of Bateni *et al.* [6]. We highlight the main contributions of our work and provide a detailed comparison of our results with [6].

---

[1] [6] only mention a $O(1)$-approximation factor. We calculate the constant to be much larger than 40.

| Problems | Results | Approximation Guarantee | Strong-oracle Queries | Weak-oracle Queries |
|---|---|---|---|---|
| $k$-means | [6] | $(O(\log^2 n \log(\frac{\log n}{\epsilon})), O(1))^1$ | $O(k^2 \log^4 n \log^2(\frac{\log n}{\epsilon}))$ | $O(nk \log^2 n \log(\frac{\log n}{\epsilon}))$ |
| | Our | $(O(\frac{\log n}{\epsilon^3}), 40(1+\epsilon))$ | $O(\frac{k^2 \log^2 n}{\epsilon^6})$ | $O(\frac{nk \log n}{\epsilon^3})$ |
| $k$-center | [6] | $14(1+\epsilon)$ | $O(k^2 \log^4 n \log^2(\frac{\log n}{\epsilon}))$ | $O(nk \log^2 n \log(\frac{\log n}{\epsilon}))$ |
| | Our | $6(1+\epsilon)$ | $O\left(k^3 \log^2 n \log \frac{\log n}{\varepsilon}\right)$ | $O\left(nk \log n \log \frac{\log n}{\varepsilon}\right)$ |

Table 1: Comparison of approximation guarantees and query complexities for $k$-means and $k$-center problems with Bateni *et al.* [6] in the weak-strong oracle model. For $k$-means, we report the bi-criteria approximation guarantees obtained by our algorithm as well as [6].

- $k$-means++ algorithm is widely used to solve $k$-means, primarily when complete distance information is known. $k$-means++ algorithm has also been extended to work in presence of outliers [7, 11], and when there are errors in the computation of the distribution [8, 16]. However, it was not known whether $k$-means++ works with unreliable distance estimates as in the weak-strong model. Recall that $k$-means++ uses a non-uniform sampling algorithm, $D^2$-sampling, to sample centers. We show that even though the distance estimates given by the weak-oracle are unreliable, it is possible to come up with a distance measure with respect to which one can run the $k$-means++ algorithm to obtain a good approximation guarantee for $k$-means. We believe one of the main contributions of this work is to adapt the $k$-means++ algorithm to work in this restricted model.

- Bateni *et al.* [6] give a constant approximation algorithm using $O(k^2 \log^4 n \log^2(\log n/\epsilon))$ strong-oracle queries and $O(nk \log^2 n \log(\log n/\epsilon))$ weak-oracle queries. Our algorithms use $O(k^2 \log^2 n/\epsilon^6)$ strong-oracle queries and $O(nk \log n/\epsilon^3)$ weak-oracle queries to give a constant approximation for $k$-means, where we also improve the constant factor.

- Our algorithm for $k$-means in the weak-strong oracle model can be generalized to obtain a $O(2^{2z})$ factor approximation in expectation for the $(k, z)$ clustering problem.

- Bateni *et al.* [6] adapts Meyerson's sketch for online facility location for $k$-means problem in the weak-strong model [22]. Their algorithm guesses the optimal value of $k$-means solution, and using a bounded aspect-ratio assumption on the input instance, they show that the number of such guesses is limited. Our algorithm doesn't use any such assumption.

- For the $k$-center problem, we obtain the following improvements over Bateni *et al.* [6]. We give a $6(1 + \epsilon)$-approximation for $k$-center using $O(k^3 \log^2 n \log \log n/\epsilon)$ strong-oracle queries and $O(nk \log n \log \log n/\epsilon)$ weak-oracle queries. This is in contrast to [6] who give a $14(1+\epsilon)$-approximation using $O(k^2 \log^4 n \log^2(\log n/\epsilon))$ strong-oracle and $O(nk \log^2 n \log(\log n/\epsilon))$ weak-oracle queries.

**Experiments** We experimentally verify the performance of our algorithms for $k$-means and $k$-center problems on synthetic as well as real-world datasets. For experiments on synthetic datasets, we use datasets generated using the stochastic block model ([24, 1]). For experiments on real-world datasets, we use MNIST dataset [10] with SVD and t-SNE embeddings [25]. We compare the experimental results of our algorithms with those of of [6], and note that our algorithms use significantly fewer number of queries to output clusterings of comparable cost. Our algorithms for $k$-means and $k$-center use at least 38% and 12% fewer queries, respectively, compared to [6]. The experimental details are provided in Section 4 and Appendix H.

## 1.3 Technical overview

In this section, we describe the main ideas of our algorithm for $k$-means in the weak-strong oracle model. One of the most widely used algorithms for $k$-means is the $k$-means++ seeding algorithm that works in $k$ iterations. In the first iteration, it chooses a point uniformly at random. Each of the remaining $(k-1)$ iterations samples a point following a non-uniform and adaptive sampling distribution known as the $D^2$ distribution that depends on the previously chosen centers. Sampling a point from the $D^2$-distribution is called $D^2$-sampling. The $k$ points sampled using $D^2$-sampling are returned as the solution for $k$-means. The $D^2$-distribution is such that the probability of choosing a point $x$ is proportional to the squared distance $d(x, C)^2$, where $C$ is a set of chosen centers

and $d(x, C) = \min_{c \in C} d(x, c)$. [2] showed a constant factor bi-criteria approximation for the oversampling version of $k$-means++ in which $O(k)$ iterations of $D^2$-sampling are used.

One of the main contributions of this work is to show that one can adapt $k$-means++ to the weak-strong oracle model. It is easy to observe that one can simulate $k$-means++ in the strong-oracle model using $O(nk)$ strong-oracle queries. However, things become complicated in the weak-strong model, in which the distances returned by the weak-oracle cannot be trusted. To construct the $D^2$-sampling distribution, one must calculate $d(x, C)$ for any point $x$ and any set $C$ of centers. Since $d(x, C) = \min_{c \in C} d(x, c)$, if we want to compute this distribution exactly, it would require $\Omega(nk)$ strong-oracle queries. Since we don't want to use too many strong-oracle queries, we proceed as follows. Using weak-oracle query answers, we compute an estimate $d_{km}^{est}(x, C)$ such that with high probability, $d(x, C)$ and $d_{km}^{est}(x, C)$ are within some small additive factor. We construct the sampling distribution to be used by the algorithm using these $d_{km}^{est}(x, C)$ values for all points $x$.

Let us see how we compute $d_{km}^{est}(x, C)$ for all $x \in X$. The main idea here is to exploit the fact that the weak-oracle distances are wrong independently with probability $\delta < 1/2$. Consider any $c \in C$. Let us obtain an estimate on $d(x, c)$ using only weak-oracle queries. Suppose there are many points in a ball $B(c, r_c)$ around center $c$ of radius $r_c$ for a reasonably small value of $r_c$. More specifically, suppose $r_c$ is such that $B(c, r_c)$ contains $\Omega(\log n)$ points. Then, the key idea is that the median of the weak-oracle distance queries on pairs $(x, y)$ with $y \in B(c, r_c)$ is a good estimate on $d(x, c)$. More formally, we use $d_{km}^{est}(x, c) = median\{\widetilde{WO}(x, y) : y \in B(c, r_c)\}$ as an estimate for $d(x, c)$. We note that [6] also uses the median of query answers to estimate $d(x, c)$. However, their algorithm used these values differently, as we will see in the remaining discussion. Since each weak-oracle query answer is wrong independently with probability at most $1/2$, using Chernoff bounds, one can show that for sufficiently large-sized ball $B(c, r_c)$, with high probability, $|d(x, c) - d_{km}^{est}(x, c)| \leq r_c$. Now, to approximate $d(x, C)$, we compute $d_{km}^{est}(x, c) + r_c$ for all $c \in C$ and set $d_{km}^{est}(x, C) = \min_{c \in C}\{d_{km}^{est}(x, c) + r_c\}$. Once we have the sampling distribution $\widetilde{D^2}$ for which a point $x$ is sampled with probability proportional to $d_{km}^{est}(x, C)^2$, we show that the approach of [2] can be adapted to obtain a constant factor bi-criteria approximation for $k$-means that succeeds with high probability. Even though the analysis becomes non-trivial and we incur an approximation loss, the high-level analysis ideas of [2] goes through.

The main idea is to run the oversampling $k$-means++ where we iteratively sample and update the center set $C$. However, the sampling is with respect to $d_{km}^{est}(x, C)^2$ instead of $d(x, C)^2$. The goal is to pick a good center from every optimal cluster. Let us see why oversampling using $d_{km}^{est}(x, C)^2$ will be sufficient to find good centers from every optimal cluster, and hence obtain a good approximation guarantee. For the simplicity of discussion, let us first assume that every optimal cluster has $\Omega(\log n)$ points. We will later see how to drop this assumption. Let us draw a parallel with oversampling $k$-means++ that samples using accurate distance values. Why does it manage to pick good centers from every optimal cluster? Consider an intermediate center set $C$. Certain optimal clusters will have a good representative center in $C$, whereas others remain 'uncovered'. Sampling using the $D^2$ distribution boosts the probability of sampling from an uncovered cluster, so there is a good chance that the next sample belongs to an uncovered cluster. Moreover, we can argue that given the next sample is from an uncovered cluster, there is a good chance that it will be a good center from that cluster. Let's see whether the same argument applies when sampling uses $d_{km}^{est}(x, C)^2$ instead of $d(x, C)^2$. If our center set $C$ has a single good center (or a few) from an optimal cluster, say $A$, can we consider the cluster to be covered? No. The probability of sampling from $A$ may remain high since the distance estimate of points in $A$ to the centers in $C \cap A$ may be completely off. When does the distance estimate start becoming tighter? This happens when there are $\Omega(\log n)$ good centers from $A$ in $C$. This is when we can call the cluster $A$ 'settled'. So, instead of 'covering' every cluster (if correct distances are known), we care about 'settling' every cluster in our current model. So, let us see if we can settle every cluster if we keep sampling centers. We can show that unless the current center set $C$ is already good, there is a good chance that the next center will be sampled from one of the unsettled optimal clusters. Further, we can also argue that conditioned on sampling from an unsettled cluster, there is a good chance that the sampled center will be a good center (i.e., reasonably close to the optimal center). So, as long as we sample sufficiently many centers, every optimal cluster will get settled with high probability. The oversampling factor is $O(\log n)$, i.e., we end up with a centre set $C$ with $O(k \log n)$ centers. Finally, we will use $d_{km}^{est}(x, C)$ to assign points $x$ to centers and create a weighted point set $C$ (the weight of a point in $C$ is the number of points assigned to it), on which a standard constant approximation algorithm using strong-oracle queries is used to find the

final set of $k$ centers. Since the set $C$ has $O(k \log n)$ points, we will need $O(k^2 \log^2 n)$ strong oracle queries to find all the interpoint distances to run the constant approximation algorithm as well as the distance estimates. Using known techniques, it can be shown that this final center set gives a constant approximation. To drop the assumption that all optimal clusters have $\Omega(\log n)$ points, we argue if an optimal cluster does not have adequate points, the oversampling procedure will sample *all* the points from that cluster with high probability, which is also a favourable case. More discussions on algorithmic ideas for the $k$-center algorithm and related results can be found in Appendix C.

### 1.4 Related works

Bateni *et al.* [6] initiate the study of clustering problems in the weak-strong oracle model and design constant approximation algorithms for $k$-means and $k$-center problems. They also prove that any constant factor approximation algorithm for $k$-means or the $k$-center problem requires $\Omega(k^2)$ strong-oracle queries in the weak-strong model. Galhotra *et al.* [13] obtain the following results for clustering problems in a related model. They design constant factor approximation algorithms for $k$-means and $k$-center clustering problems given access to a *quadruplet oracle* that takes two pairs $(x_1, y_1)$ and $(x_2, y_2)$ of points as input and returns whether the pairwise distances are similar or far from each other. A relatively detailed discussion on related works can be found in Appendix B.

## 2  Algorithms for $k$-means in weak-strong oracle model

In this section we design a constant factor bi-criteria approximate solution for $k$-means in the weak-strong oracle model with the assumption that all optimal clusters have size at least $480 \log n/\epsilon$. We remove this assumption later. We prove the following result.

**Theorem 2.1.** *Let $\epsilon \in (0,1)$, $\delta \leq 1/3$. There exists a randomized algorithm for $k$-means that makes $O(k^2 \log^2 n/\epsilon^4)$ strong-oracle queries and $O(nk \log n/\epsilon^2)$ weak-oracle queries to give a $\left( O(\log n/\epsilon^2), 40(1+\epsilon) \right)$ bi-criteria approximation for $k$-means and succeeds with constant probability, assuming every optimal cluster has size at least $480 \log n/\epsilon$.*

### 2.1  Algorithm and analysis

Since the weak-oracle gives a wrong answer to a query independently with probability $\delta < 1/2$, we define an alternate distance measure between points and a set of centers that we use in our algorithm. Let $C$ denote a set of centers. For any $x \in X$ and any $c \in C$, we query for the distance between $x$ and $c$ to the weak oracle $\widetilde{WO}$ to obtain a possibly wrong answer $\widetilde{WO}(x,c)$. For any fixed $x$, we query the weak-oracle for each center $c$ in $C$, and use these query answers $\widetilde{WO}(x,c)$ to come up with an upper bound on the distance between a point $x$ and a set $C$ of centers defined as follows. We use $\delta \leq 1/3$ for this discussion.

**Definition 2.1.** *Let $x$ and $y$ be any two points in $X$ and we want to estimate the distance between $x$ and $y$. Let radius $r_y \geq 0$ be such that the ball $B(y, r_y)$ contains at least $180 \log n$ points. Then, for any $x \in X$, we define the distance from $x$ to $y$ as $d_{km}^{est}(x,y) = median\{\widetilde{WO}(x,z) | z \in B(y, r_y)\}$.*

We make the following claim with respect to the above distance measure. A similar lemma was proved in [6]. The proof can be found in the Appendix.

**Lemma 2.1.** *Following Definition 2.1, for any $x \in X$, with probability at least $\left(1 - 1/n^5\right)$, we have $|d_{km}^{est}(x,y) - d(x,y)| \leq r_y$.*

We use the above definition to come up with an upper bound on the distance between a point $x$ and a set $C$ of centers. For any $c \in C$, let $r_c$ denote a radius such that $B(c, r_c)$ has at least $180 \log n$ points inside it. We use Definition 2.1 to come up with an upper bound on the distance between $x$ and any $c \in C$ as $d_{km}^{est}(x,c) + r_c$. Following Lemma 2.1, with high probability, this upper bound holds. That is, with high probability, $d(x,c) \leq d_{km}^{est}(x,c) + r_c$. Using an union bound over all $c \in C$, all these upper bounds hold and hence, the minimum of these upper bounds would also hold. This motivates the following definition on the distance between a point $x$ and a set $C$ of centers.

**Definition 2.2.** *Let $x$ be any point in $X$ and let $C$ be a set of centers of size at least $180 \log n$. For any $c \in C$, let $r_c$ denote a radius value such that the ball $B(c, r_c)$ contains at least $180 \log n$ points. We define the distance between $x$ and $C$ as $d_{km}^{est}(x,C) = \min_{c \in C}\{d_{km}^{est}(x,c) + r_c\}$.*

We state the following lemma with respect to the above distance measure and prove it in Appendix.

**Lemma 2.2.** *Let $x$ be any point in $X$ and let $C$ denote a set of centers of size at least $180 \log n$. Then, with probability at least $(1 - 1/n^4)$, $d(x, C) \leq d_{km}^{est}(x, C)$. In other words, with probability at least $(1 - 1/n^4)$, there exists a center $c \in C$ within a distance of $d_{km}^{est}(x, C)$ from $x$.*

**Estimating $d_{km}^{est}(x, C)$ in weak-strong oracle model**  Next, we describe how we estimate $d_{km}^{est}(x, C)$ for all $x \in X$ in the weak-strong oracle model. We assume that there are at least $180 \log n$ centers in $C$. Consider any $c \in C$. We query the strong-oracle $SO$ with all pairs $c_i, c_j \in C$ to obtain the exact distances between centers $c_i$ and $c_j$ as $SO(c_i, c_j)$. For each $c \in C$, we find the smallest $r_c$ such that $B(c, r_c)$ contains at least $180 \log n$ points in $C$. Consider any $x \in X$ for which we want to estimate $d_{km}^{est}(x, C)$. We query the weak-oracle with $x$ and $y \in B(c, r_c)$ to obtain $\widetilde{WO}(x, y)$ for all $y \in B(c, r_c)$, and use these weak-oracle query answers to compute $d_{km}^{est}(x, c)$ following Definition 2.1. We compute $d_{km}^{est}(x, c)$ for all $c \in C$. Finally, to compute $d_{km}^{est}(x, C)$, we find the minimum over all $c \in C$ of $d_{km}^{est}(x, c) + r_c$, as mentioned in Definition 2.2.

---

**Algorithm 1:** Algorithm for $k$-means in weak-strong oracle model

**Input**  : Dataset $X$ and an integer $k > 0$, $\epsilon \in (0, 1)$, $\delta = 1/3$.
**Output** : A set of $O(k \log n / \epsilon^2)$ centers and an assignment of $x \in X$ to centers.
Let $C_1$ be a set of $180 \log n$ points chosen arbitrarily from $X$.      /* `Initial centers` */
Set $t = 4320 \cdot 800000/\epsilon^2 \cdot k \log n$.
**for** $i = 1$ **to** $t$ **do**
  For each $x \in X$, compute $d_{km}^{est}(x, C_i)$ following Definition 2.2.
  Compute distribution $\widetilde{D^2}$ that samples $x \in X$ with probability proportional to $d_{km}^{est}(x, C_i)^2$.
  Sample a point $s_i \in X$ using distribution $\widetilde{D^2}$.          /* `Sample new centers` */
  Make strong-oracle queries $SO(s_i, c)$ for all $c \in C_i$.
  Update $C_{i+1} \leftarrow C_i \cup \{s_i\}$.
**end**
Let $\{c_1, c_2, \ldots, c_h\}$ be an arbitrary ordering of the centers in $C_{t+1}$, where $h = 180 \log n + t$.
Initialize weights $w(c_i) = 0$ for $i \in [h]$.
**for** $x \in X$ **do**
  Compute $d_{km}^{est}(x, C_{t+1})$ and find $c_x$ for which minimum is achieved in Definition 2.2.
  Assign $x$ to $c_x$.
  Update $w(c_x) \leftarrow w(c_x) + 1$.                 /* `Construct weighted instance` */
**end**
**return** $C_{t+1}$ *and assignment of points in $x \in X$ to centers in $C_{t+1}$ as determined above*

---

We give a high-level description of Algorithm 1. Algorithm 1 starts with a set $C$ of $180 \log n$ centers chosen from $X$, and computes $d_{km}^{est}(x, C)$ for all $x$. It constructs the $\widetilde{D^2}$-sampling distribution for which a point $x \in X$ is sampled with probability proportional to $d_{km}^{est}(x, C)^2$. Algorithm 1 samples a center in each of $t$ iterations and updates the center set $C$, and once the $t$ rounds of sampling are done, it constructs a weighted instance. Finally, it returns this set of centers with an assignment of points to these centers. The solution for $k$-means is obtained using a constant approximation algorithm on this weighted instance. The detailed analysis of the algorithm is given in Appendix E.

## 3  Algorithms for $k$-center in weak-strong oracle model

We design a randomized algorithm in the weak-strong oracle model that gives a $6(1 + \varepsilon)$-approximation for the $k$-center problem. The main idea of the algorithm can be better understood in a simpler setting where accurate distances are known. Let us see an outline of this algorithm. We assume that the optimal $k$-center radius $r_{opt}$ is known. This is not an unreasonable assumption since we can guess the optimal radius within a $(1 \pm \varepsilon)$-factor by iterating over discrete choices of the radius. The algorithm is a simple "*greedy ball-carving*" procedure that is commonly used in the context of the $k$-center problem and can be stated as follows (also stated as Algorithm 3 in [6]): *While all points are not removed, pick a centre $c$ and remove (carve-out) all points within $2r_{opt}$ of $c$.* We can show that this procedure outputs at most $k$ centers that is a 2-approximate solution to the $k$-center problem.

Our $6(1 + \varepsilon)$-approximation algorithm follows the above template. The only complication is that during the ball carving step, where we remove all points within the radius $2r_{opt}$, we need accurate distances, which we do not have in the weak-strong oracle model. Let us continue assuming that the optimal radius $r_{opt}$ is known. We will give a 6-approximation under this assumption, which changes to $6(1 + \varepsilon)$ when the assumption is dropped. To enable distance estimates, along with a center $c_i$, we must also pick a set of nearby points to $c_i$, which will help in estimating the distance of any point $y$ to $c_i$ following Lemma 2.1. A reasonable way to do this would be to uniformly sample a set $T_i$ of points (instead of one), use the strong-oracle queries on the subset and pick a center $c_i \in T_i$ with the largest number of points in $T_i$ within a radius $2r_{opt}$. Let $S_{c_i}$ denote the subset of points in $T_i$ within a distance of $2r_{opt}$ to $c_i$. We can now obtain distance estimates using the tuple $(c_i, S_{c_i})$ using the weak-oracle queries as in Lemma 2.1 and carve out those points and assign them to $c_i$ that satisfy $d_{km}^{est}(c_i, z) \leq r$ for an appropriate value of $r$. Let $c_i$ belong to the optimal cluster $X_i^*$. What should the appropriate value of $r$ be such that the points in $S \cap X_i^*$ are guaranteed to get carved out and assigned to $c_i$? Since the distance estimates are inaccurate, we must use $r = 4r_{opt}$. However, this may cause a point that is $6r_{opt}$ away from $c_i$ to get carved out and assigned to $c_i$. This is the reason we obtain an approximation guarantee of 6. This high-level analysis lacks two relevant details: (i) how do we eliminate the assumption that $r_{opt}$ is known, and (ii) probability analysis for all points being assigned a center within distance $6r_{opt}$. We remove the assumption about $r_{opt}$ being known by iterating over discrete choices $Rad = (1 + \varepsilon)^0, (1 + \varepsilon), (1 + \varepsilon)^2, \ldots$ and we argue that the ball carving succeeds for $Rad = r_{opt}(1 + \varepsilon)$ with high probability. Hence, the approximation guarantee becomes $6(1 + \varepsilon)$. Instead of iterating over the possible choices of $Rad$ linearly, we can do that using binary search, which results in the running time and query complexity getting multiplied by a factor of $O(\log \log \Delta / \varepsilon)$, where $\Delta$ is the aspect ratio. Assuming $\Delta$ to be polynomially bounded, this factor becomes $O(\log \log n / \varepsilon)$. We describe our $k$-center algorithm as Algorithm 2 and state the theorem that we prove in Appendix G.

---

**Algorithm 2:** `Weak-Greedy Ball Carving`

---

**Input** : Set of points $S$, radius $Rad$
**Output :** A set of $C = \{c_1, \ldots, c_m\}$ centers and assignment of points in $S$ to centers in $C$
Initialize: $C = \{\}$
**while** *$S$ is not empty* **do**

> If $(|C| = k)$ **abort**
> If $(|S| \leq 180k \log n)$, use strong-oracle queries to find remaining centers covering all points in $S$ within distance $2Rad$. If this is not possible, **abort**. Otherwise, output $k$ centers.
> Pick a subset $T \subset S$ of size $180k \log n$ uniformly at random
> Query the strong-oracle to find distances between elements in $T$ and use these to find a center $c \in T$ such that $|S_c|$ is maximised, where $S_c \equiv |\mathcal{B}(c, 2Rad) \cap T|$.
> If $(|S_c| < 180 \log n)$ **abort**
> Retain $180 \log n$ points in $S_c$ (*i.e., remove the extra point*)
> Assign any point $s \in S$ to center $c$ if $d_{km}^{est}(s, c) \leq 4Rad$      /* Recover cluster */
> Add $c$ to $C$ and remove all points assigned to $c$ from $S$.

**end**
**return** *$C$ and the assignments*

---

**Theorem 3.1.** *There exists a $6(1 + \epsilon)$-approximation algorithm for the $k$-center problem that makes $O\left(k^3 \log^2 n \log \frac{\log n}{\varepsilon}\right)$ strong-oracle and $O\left(nk \log n \log \frac{\log n}{\varepsilon}\right)$ weak-oracle queries and succeeds with probability at least $(1 - 1/n^4)^2$.*

## 4   Experimental Results

We run our algorithms for $k$-means and $k$-center problems on synthetic as well as real-world datasets to demonstrate that our algorithms can be implemented efficiently in practice, and provide better results compared to Bateni *et al.* [6]. Here, we provide experimental results only for $k$-means on synthetic data. The experiments were conducted on a server with 1.5 TB RAM and 64 CPU cores. More details about the experiments including results for $k$-means on MNIST dataset and experimental results for $k$-center are given in Appendix H.

**Remark 4.1.** *To compare our results with Bateni et al. [6], we report the percentage of point queries used by the algorithm in our experimental results, where a point query gives an embedding of a point. This is in contrast to a strong-oracle query that gives the exact distance between two points.*

**Datasets**   We run our algorithms on synthetic as well as real-world datasets. We generate synthetic data using the Stochastic Block Model (SBM) ([24, 1]). We use three synthetic datasets in our experiments with the number of points $n$ being $10k, 20k$ and 50k. The number of clusters $k$ is 7 and the points belong to a 7-dimensional space. The points in the $i$th cluster of the datasets are drawn from a Gaussian distribution $N(\mu^{(i)}, I)$, where $\mu^{(i)}[i] = 10^5$ and $\mu^{(i)}[j] = 0$ for $j \neq i$.

**Construction of the weak-oracle**   We create perturbed distance matrices $M$ for the datasets using which the weak-oracles answer queries. These matrices are initialized with actual distances between points in the datasets. For the SBM-based dataset, for any two points $i, j$, independently with probability $\delta$, $M(i,j) = 10^5$ if $i$ and $j$ belong to the same cluster and $M(i,j) = 1$ otherwise.

**Baselines in our experiments**   Running $k$-means++ with the strong-oracle distances gives us the *strong-baseline* for our experiments. We calculate the approximation factors by computing the ratio of the cost of the algorithm with this strong baseline.

**Experimental results on $k$-means clustering**   We run our $k$-means algorithm on SBM-based synthetic datasets of size $n = 10k, 20k$ and $50k$ where the failure probability $\delta$ of the weak-oracle is set to be $0.1, 0.2$ and $0.3$. For each of the above choices, we run our algorithm on the datasets with varied number of point queries. Table 2 shows our experimental results on $k$-means clustering on SBM-based datasets of size $n = 10k, 20k$ and $50k$. We conducted experiments for different combinations of $\delta$ and the number of point queries. In Table 2, we give the values for which the value of (number of point queries $\times \log$(cost of clustering)) is minimized. We report variance figures in Appendix H.

| $n$ | % of point queries | | | Approximation factor | | |
|---|---|---|---|---|---|---|
|  | $\delta = 0.1$ | $\delta = 0.2$ | $\delta = 0.3$ | $\delta = 0.1$ | $\delta = 0.2$ | $\delta = 0.3$ |
| $10k$ | 0.77 | 0.79 | 0.79 | 0.38 | 0.388 | 0.379 |
| $20k$ | 0.045 | 0.047 | 0.0495 | 0.386 | 0.379 | 0.3769 |
| $50k$ | 0.02 | 0.021 | 0.0214 | 0.3819 | 0.3727 | 0.3842 |

Table 2: Performance of our $k$-means algorithm on SBM-based synthetic data

**Comparison with Bateni** *et al.* **[6]**: Figures 1 and 2 capture how $k$-means clustering cost varies with the number of point queries when these algorithms are run on SBM-based synthetic datasets of size $10k$ and $20k$. The blue, green and red lines represent results for $\delta = 0.1$, $\delta = 0.2$, and $\delta = 0.3$, respectively. The solid lines represent the performance of our algorithm while the dashed lines represent results from [6]. Bateni *et al.* [6] reported these values as a plot. In order to compare our results with [6], we interpreted the values from those plots.

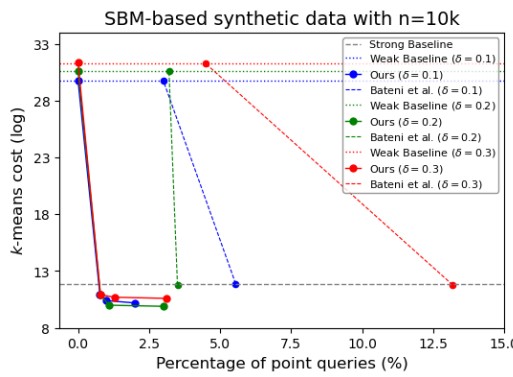

Figure 1: $k$-means on SBM dataset with $n = 10k$

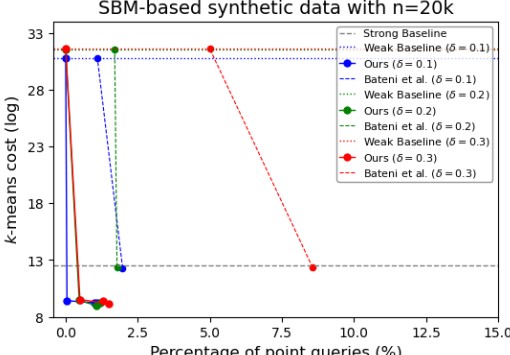

Figure 2: $k$-means on SBM dataset with $n = 20k$

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
