# OpenReview forum: "Improved Algorithms for Clustering with Distance Oracles"
_NeurIPS.cc/2025/Conference — Submitted to NeurIPS 2025_

### Official Review · Reviewer_q3eE · 2025-06-20

**Clarity:** 3
**Significance:** 2
**Originality:** 3
**Rating:** 4
**Confidence:** 3

**Summary:**

The submission proposed novel algorithms for k-center and k-means clustering in a model which aims at minimizing the amount of information required about the data points. In particular, the model considers both "strong" queries (which provide exact information but should be minimized), and "weak" queries (which provide potentially faulty information).

**Questions:**

The authors are welcome to respond to any of the points raised in the "Strengths and Weaknesses".

**Ethical Concerns:**

["NO or VERY MINOR ethics concerns only"]

**Final Justification:**

See response to authors.

**Limitations:**

yes

**Paper Formatting Concerns:**

None.

**Quality:**

3

**Strengths And Weaknesses:**

The submission is well-written overall. Its contribution primarily lies on the foundational side, but supports its findings with experimental evaluations. I especially appreciated the inclusion of the "comparison with known results" and "Technical overview" subsections/paragraphs - these are often missing in NeurIPS submissions. Nevertheless, I do have some concerns about the submissions' presentation and contributions, and at present these are preventing me from more strongly suggesting acceptance.

[Minor] The messaging regarding the main results is somewhat confusing. The Main Results section begins by discussing results for the Strong-oracle, but as far as I understand these are all direct observations and the non-trivial contributions all apply to the weak-strong oracle model.

[Medium] The model assumes that weak queries have a probability of returning an entirely random value, and that these values are persistent (i.e., that repeating the same query will not provide different results). For the latter, I understand that without persistence one could repeat the query multiple times in order to circumvent the whole idea behind the model - but nevertheless, I wonder whether the assumption can be justified or motivated by real-world applications. For the former, I would have expected that the weak queries simply provide answers which have a probability distribution around the real hidden values... again, I understand that this is different, but there seems to be no justification for why the proposed model is better suited for at least some applications.

Note that I am aware that the preceding work [6] made the same assumptions, but nevertheless the model should be justified here as well (especially since here the authors "double down" on these assumptions with their main contributions). Moreover, in my quick parse of [6] I didn't spot a careful justification of these assumptions there either.

[Minor] The current explanation of k-means in subsection 1.1 does not make it clear that the k "centers" are points from the metric space and not elements of X. I suggest replacing "centers" on the 4th row of 1.1 with "points in \mathcal{X} (called \emph{centers})".

[Suggestion] At the top of page 3, it would be nice to mention the constant in "constant factor bi-criteria approximation".

[Minor] The wall of text on page 5 could be improved, e.g., by introducing a new paragraph at "Let’s see whether the same argument applies..."

[Suggestion] The justification for why the proposed research should appear at ICLR could be strengthened by outlining that clustering of unknown/incomplete data has been studied by the machine learning research community in a variety of settings. For instance, there are several algorithmic-complexity papers on clustering incomplete data that could be mentioned:

--Tomohiro Koana, Vincent Froese, Rolf Niedermeier: The complexity of binary matrix completion under diameter constraints. J. Comput. Syst. Sci. 132: 45-67 (2023)

--Robert Ganian, Thekla Hamm, Viktoriia Korchemna, Karolina Okrasa, Kirill Simonov: The Complexity of k-Means Clustering when Little is Known. ICML 2022: 6960-6987

--Eduard Eiben, Fedor V. Fomin, Petr A. Golovach, William Lochet, Fahad Panolan, Kirill Simonov:
EPTAS for k-means Clustering of Affine Subspaces. SODA 2021: 2649-2659

--Eduard Eiben, Robert Ganian, Iyad Kanj, Sebastian Ordyniak, Stefan Szeider: The Parameterized Complexity of Clustering Incomplete Data. AAAI 2021: 7296-7304

[Minor] The formulation of Theorem 1.1 is too vague: it doesn't clarify what the impact of epsilon is (whether it affects the runtime, the approximation factor or both only becomes clear later). Overall, I'd suggest stating the result in  a way which is explicit about the runtime and factor, similarly as Theorem 1.2.

[Comment] On the technical side, the results are non-trivial and of sufficient technical depth to justify presentation at NeurIPS. However, the presentation of the results and comparison to the direct predecessor [6] fails to mention that the improvements only target a rather specific (and small) subset of the scope of [6]. In this sense, the contributions can be seen as somewhat narrow and comparatively weaker (but perhaps still of sufficient interest to consider acceptance).

[Major] The preceding work [6] introduces and considers strong point queries as well as strong distance queries. In fact, the main contributions of [6] seem to target strong point queries first and foremost - for instance, these are what the experiments in [6] focus on. The submission however does not mention this at all in the discussion and comparison to [6]; everything up to the experimental section focuses on strong distance queries, and then strong point queries are abruptly introduced as a side note when they needed in the experimental section. I am wondering:
1) Do the results obtained here also yield improvements to the number of strong point queries required (Theorems 1 and 2 in [6]), or are the techniques exclusive to strong distance queries?
2) How did you translate theoretical algorithms that consider strong distance queries to experimental evaluations that count the number of strong point queries, and is this explained in the submission?

---

> ### Author Rebuttal · Authors · 2025-07-28
>
> We thank the reviewer for asking really interesting questions. Due to space constraints, we could not address the minor points raised by the reviewer. We will definitely take care of them in the final version.
>
> (**A**): _Justification of the weak-strong oracle model:_
>
> First, let us note that the weak-oracle model is an attempt to capture the limited distance information between points that one often has to tackle to solve clustering problems in practice. To the best of our knowledge, there have not been many works in this context, and we think the weak-oracle model, despite its shortcomings, is a good model to start with. That said, the weak-oracle model and its variants, as discussed below, should be useful in some applications.
>
> In the weak-oracle model, it is assumed that the weak-oracle returns the exact distance between any two points independently with probability $(1-\delta)$ and with the remaining probability, it can report any arbitrary value. We first note that one can relax the requirement that the weak-oracle needs to report the exact distance between two points as follows. The weak-oracle could return a value within a constant factor of the actual distance, with high probability. Our algorithms work with this modified weak-oracle and will give a solution of quality at most a constant factor worse compared to the earlier solution. Regarding the independence assumption of the weak-oracle, we note that our algorithms should also work in bounded independence models. We apply Chernoff-Hoeffding under independence, which can be replaced with other concentration inequalities under bounded independence.  However, the algorithms do not work if the query answers are arbitrarily correlated. We are not aware of many studies that design algorithms for clustering problems that work with arbitrarily correlated Oracle answers. One recent work along these lines is ``MAC Advice for Facility Location Mechanism Design'' by Barak et al. (Neurips'24), which designs mechanisms for facility location problems given noisy Oracle access and highlights potential difficulties in obtaining strong algorithmic results without the independence assumption. On the other hand, there are applications where assuming that the errors are independent is not completely unreasonable, such as when distances are obtained using a crowd-sourced platform. So, the independence assumption provides a good initial platform for understanding important questions, such as whether it is possible to retain the approximation guarantees of algorithms such as k-means++ (for k-means) and ball-carving (for k-center) in this model.
>
> (**B**): _Whether weak-oracles can be simulated by sampling distances from probability distributions:_
>
> We find the idea of sampling weak-oracle distances from probability distributions to be very appealing, and we thank the reviewer for the suggestion. Here is what we observe about the proposed model. First, consider the scenario in which the query answers are not persistent. In this case, one can ask the same query multiple times and use concentration bounds for the queried distances. This makes the problem easy. Now, let us assume that the query answers are persistent. If the probability distributions have strong concentration properties, then with high probability (say, with probability $(1-\delta)$), the samples will be close to the expected value and with remaining probability (with probability at most $\delta)$, the sample will be far from the expected value. We believe that this can be modelled in the weak-strong oracle model that we consider in this paper with an appropriate choice of parameters. Note that one can relax the weak-oracle to require that it only returns an answer within a constant factor of the actual value, and our algorithms with this modified oracle will return solutions that are at most a constant factor worse. The above discussion assumes that the distances are drawn from independent probability distributions and that the sampling is also done independently.
>
> (**C**): _Discussion on use of point queries and edge (pair) queries:_
>
> In this paper, we chose to work with only weak and strong edge queries (or pair queries) in the weak-strong oracle model. We believe that assuming access to point queries makes a stronger assumption on the model. To recall, a point query gives the complete embedding of a point, whereas an edge query only gives the distance between the two points. For any set of $t$ points, one can make either $t$ point queries or ${t\choose 2}$ edge queries to obtain the complete distance information between the points. We highlight that it is possible to rewrite our algorithms to use point queries instead of edge queries, and these algorithms would use fewer point queries compared to the algorithms of Bateni et al.
>
> In order to compare the empirical performances of our algorithms with those of Bateni et al., we ran our algorithms that use edge queries, but while reporting the results, we only mentioned the number of point queries used by our algorithms. We find the corresponding numbers of point queries of Bateni et al. either from their tables or by interpreting their plots. We mentioned this in Remark 4.1, and at places where we compared our results with Bateni et al..
>
> (**D**): _Experimental results using edge queries:_
>
> Our algorithms for both k-means and k-center sample points of the input, and use strong-edge queries to obtain exact distances between the sampled points. Had we used point queries in our algorithms, the same guarantees could be achieved by making a point query for each sampled point. We make strong-edge queries to obtain distances between any two sampled points, whereas Bateni et al. make a point query for each sampled point. Let $t_1$ and $t_2$ denote the number of points sampled by our algorithm and Bateni et al., respectively. Then, our algorithm makes ${t_1\choose 2}$ strong-edge queries, whereas Bateni et al. make $t_2$ point queries.
>
> Since some reviewers were confused by our use of strong edge queries in the descriptions of algorithms and the use of point queries to state experimental results, we decided to report our experimental results using the number of strong edge queries used by the algorithms. We also calculate the number of strong edge queries by Bateni et al. and compare it with our results.
>
> Our algorithms use substantially fewer strong edge queries compared to those of Bateni et al. We report the percentage of strong edge queries used by our algorithms and those of Bateni et al. for both $k$-means and $k$-center problems for both synthetic and real-world datasets, and for different choices of the failure probability $\delta$. For the $k$-means problem on SBM-based synthetic datasets of sizes $n = 10k, 20k, 50k$, our algorithms reduce the number of strong edge queries by at least $93$ percent compared to Bateni et al. Our algorithm for $k$-means on the MNIST dataset using SVD and $t$-SNE embeddings improves the query complexity by at least $61$ and $99$ percent, respectively.
>
>
> |   n   |   **% of strong edge queries**  |      |    | **% of strong edge queries** |     |    |  **Improvement (%)**     |      |       |
> |:-----:|:-----------:|:----------:|:----------:|:--------------:|:----------------:|:----------------:|:-------------:|:------:|:-------:|
> |           |   **(Bateni et al.)** |         |         |  **(Ours)** |        |           |        |      |       |
> |       |  $\delta$ = 0.1  |  &nbsp; $\delta$ = 0.2 &nbsp;  |   &nbsp; $\delta$ = 0.3 &nbsp;  |  &nbsp; $\delta$ = 0.1 &nbsp;  |   &nbsp; $\delta$ = 0.2 &nbsp;  |  &nbsp; $\delta$ = 0.3 &nbsp; | $\delta$ = 0.1 | &nbsp; $\delta$ = 0.2 &nbsp;      | &nbsp; $\delta$ = 0.3 &nbsp;  |
> | 10k   |   0.1537  |  0.0614 |  0.8692  | 0.0029  | 0.0030  |  0.0030  | 98 |  94  | 99 |
> | 20k   |   0.0389  |  0.0315 |  0.7340  | 0.0020  | 0.0021  |  0.0024  | 94 | 93 | 99 |
> | 50k   |   0.0268  |  0.0167 |  0.1370  |  0.0009 | 0.0010  |  0.0011  | 96 |93 | 99  |
>
> **Table 1: Performance of $k$-means algorithms on SBM-based synthetic datasets**
>
> |   MNIST   |   **% of strong edge queries**  |      |    | **% of strong edge queries** |     |    |  **Improvement (%)**     |      |       |
> |:-----:|:-----------:|:----------:|:----------:|:--------------:|:--------------:|:--------------:|:-------------:|:------:|:-------:|
> |           |   **(Bateni et al.)** |         |         |  **(Ours)** |        |           |        |      |       |
> |       |  $\delta$ = 0.1  |  &nbsp; $\delta$ = 0.2 &nbsp;  |   &nbsp; $\delta$ = 0.3 &nbsp;  |  &nbsp; $\delta$ = 0.1 &nbsp;  |   &nbsp; $\delta$ = 0.2 &nbsp;  |  &nbsp; $\delta$ = 0.3 &nbsp; | $\delta$ = 0.1 | &nbsp; $\delta$ = 0.2 &nbsp;  | &nbsp; $\delta$ = 0.3 &nbsp;  |
> |  SVD   |  0.0018 | 0.0029 | 0.0019 | 0.0004 | 0.0005 | 0.0007 |  78| 79 | 61 |
> | t-SNE |  0.629 | 0.6263 | 1.3144 | 0.0016 | 0.0023 | 0.0026 | 99 | 99 | 99 |
>
> **Table 2: Performance of $k$-means algorithms on MNIST datasets**
>
> |   n   |   **% of strong edge queries**  |      |    | **% of strong edge queries** |     |    |  **Improvement (%)**     |      |       |
> |:-----:|:-----------:|:----------:|:----------:|:--------------:|:----------------:|:----------------:|:-------------:|:------:|:-------:|
> |           |   **(Bateni et al.)** |         |         |  **(Ours)** |        |           |        |      |       |
> |       |  $\delta$ = 0.1  |  &nbsp; $\delta$ = 0.2 &nbsp;  |   &nbsp; $\delta$ = 0.3 &nbsp;  |  &nbsp; $\delta$ = 0.1 &nbsp;  |   &nbsp; $\delta$ = 0.2 &nbsp;  |  &nbsp; $\delta$ = 0.3 &nbsp; | $\delta$ = 0.1 | &nbsp; $\delta$ = 0.2 &nbsp;      | &nbsp; $\delta$ = 0.3 &nbsp;  |
> | 10k   | 0.2115 | 0.2749 |0.2749  | 0.0199 | 0.0311 | 0.743 | 90 | 88 | 72 |
> | 20k   | 0.158 | 0.0383 | 0.2978 | 0.0120 | 0.0195 |0.0439 | 23 | 49 | 85 |
> | 50k   |  0.0158 |0.0549 | 0.1191 | 0.0048 | 0.0089 | 0.0202 | 69 | 83 |83 |
>
> **Table 3: Performance of $k$-center algorithms on SBM-based synthetic datasets**

---

> > ### Comment · Reviewer_q3eE · 2025-08-04
> >
> > Thank you for your responses - especially the further discussion of the properties of the model and a proper comparison to the previous work. I think a discussion of these aspects should have been included in the original submission. Overall, I see this as a decent paper that is above the acceptance threshold for NeurIPS, but find it difficult to raise my score further.

---

### Official Review · Reviewer_pEt7 · 2025-06-28

**Clarity:** 4
**Significance:** 3
**Originality:** 3
**Rating:** 5
**Confidence:** 4

**Summary:**

The paper designs new algorithms for clustering problems in the weak-strong distance oracle model, where the distance between points is available only via a distance oracle that offers two types of queries: a strong query always returns the true distance, and a weak query that with probability $\delta=1/3$ returns an arbitrarily corrupted value. This model describes scenarios where strong queries are computationally expensive, e.g. hard to apply or implement, hence the main measure of algorithmic efficiency is the number of strong queries.

This model was introduced in [6], which provided approximation algorithms for a few problems. The submission achieves improved approximation factor for two clustering problems, k-means and k-center, with slightly better efficiency, i.e., number of strong queries but also of weak queries. The new k-means algorithm is based on the well-known k-means++ algorithm, a very different approach from [6] which is based on Meyerson's online facility location. The algorithms' effectiveness is demonstrated also via an empirical evaluation using synthetic and real-life datasets.

**Questions:**

none

**Ethical Concerns:**

["NO or VERY MINOR ethics concerns only"]

**Final Justification:**

I still feel that the model is nice mathematically, but the paper fails to provide a solid justification. Anyway, the direction is important and thus confirm my score.

**Limitations:**

yes

**Quality:**

3

**Strengths And Weaknesses:**

The main strength is the theoretical improvement in approximation factor and number of strong/weak queries. Moreover, the k-means is based on an algorithm that is famous also as practical heuristic, hence the submission tells us how to adapt this common heuristic to use two types of queries.

Another strength is the model itself, which is abstracts nicely having access to two different distance oracle. The model was proposed in prior work (not here), but it is very timely, as access to expensive models (deep networks) is becoming prevalent, and the strong-weak model offers and examines the tradeoff between cost and accuracy.

The main weakness I see is that the model is not fully justified e.g. what practical scenarios it models reasonably well, particularly for the true distance being returned with probability $1-\delta$? It is more reasonably for a noisy physical measurement than for a deep network. Moreover, this is one of two models introduced in [6], and plausibly one that is the less relevant for deep network representation.

A second weakness is that the empirical evaluation is not very convincing, roughly for the same issues. The experiments with synthetic datasets have limited value, because the SBM is random but has some structure, and because the experiments use a specific corruption pattern, which is not necessarily hard or adversarial, and the consequence is that table 2 reports various numbers but there is no lesson to be learned. In addition, I see improvement in comparison with [6], however I could not understand what is measured here and how it's related to the theoretical part (remark 4.1), perhaps it corresponds to the other model introduced in [6]. I am ignoring the experiments with real-life datasets, because they appear only in the appendix.

Overall, I feel the strengths overweigh the weaknesses.

---

> ### Author Rebuttal · Authors · 2025-07-28
>
> We thank the reviewer for making insightful comments about our work. Please find below our replies to the queries.
>
> (**A**): _Justification of weak-strong oracle model:_
>
> First, let us note that the weak-oracle model is an attempt to capture the limited distance information between points that one often has to tackle to solve clustering problems in practice. To the best of our knowledge, there have not been many works in this context, and we think the weak-oracle model, despite its shortcomings, is a good model to start with. That said, the weak-oracle model and its variants, as discussed below, should be useful in some applications.
>
> In the weak-oracle model, it is assumed that the weak-oracle returns the exact distance between any two points independently with probability $(1-\delta)$ and with the remaining probability, it can report any arbitrary value. We first note that one can relax the requirement that the weak-oracle needs to report the exact distance between two points as follows. The weak-oracle could return a value within a constant factor of the actual distance, with high probability. Our algorithms would work with this modified weak-oracle and will give a solution of quality at most a constant factor worse compared to the earlier solution. Regarding the independence assumption of the weak-oracle, we note that our algorithms should also work in bounded independence models. We apply Chernoff-Hoeffding under independence, which can be replaced with other concentration inequalities under bounded independence.  However, the algorithms do not work if the query answers are arbitrarily correlated. We are not aware of many studies that design algorithms for clustering problems that work with arbitrarily correlated Oracle answers. One recent work along these lines is ``MAC Advice for Facility Location Mechanism Design'' by Barak et al. (Neurips 2024), which designs mechanisms for facility location problems given noisy Oracle access (oracle gives noisy locations for the clients) and highlights potential difficulties in obtaining strong algorithmic results without the independence assumption. On the other hand, there are applications where assuming that the errors are independent is not completely unreasonable, such as when distances are obtained using a crowd-sourced platform. So, the independence assumption provides a good initial platform for understanding important questions, such as whether it is possible to retain the approximation guarantees of widely used algorithms such as k-means++ (for k-means) and ball-carving (for k-center) in this model.
>
> (**B**): _Discussion on the use of point queries and edge (pair) queries:_
>
> In this paper, we chose to work with only weak and strong edge queries (or pair queries) in the weak-strong oracle model. We believe that assuming access to point queries makes a stronger assumption on the model. To recall, a point query gives the complete embedding of a point, whereas an edge query only gives the distance between the two points. For any set of $t$ points, one can make either $t$ point queries or ${t\choose 2}$ edge queries to obtain the complete distance information between the points. We highlight that it is possible to rewrite our algorithms to use point queries instead of edge queries, and these algorithms would use fewer point queries compared to the algorithms of Bateni et al.
>
> (**C**): _Experimental results using edge queries:_
>
> Our algorithms for both $k$-means and $k$-center sample points of the input, and use strong-edge queries to obtain exact distances between the sampled points. Had we used point queries in our algorithms, the same guarantees could be achieved by making a point query for each sampled point. We make strong-edge queries to obtain distances between any two sampled points, whereas Bateni et al. makes a point query for each sampled point. Let $t_1$ and $t_2$ denote the number of points sampled by our algorithm and Bateni et al., respectively. Then, our algorithm makes ${t_1\choose 2}$ strong-edge queries, whereas Bateni et al. make $t_2$ point queries.
>
> Since some reviewers became confused with our use of strong edge queries in the descriptions of algorithms, and the use of point queries to state experimental results, we decided to report our experimental results using the number of strong edge queries used by the algorithms. Since Bateni et al. only report the number of point queries, we calculate the equivalent number of strong edge queries and compare them with our results.
>
>
> Our algorithms use substantially fewer strong edge queries compared to algorithms of Bateni et al. We report the percentage of strong edge queries used by our algorithms and those of Bateni et al. for both $k$-means and $k$-center problems for both synthetic and real-world datasets, and for different choices of the failure probability $\delta$. For the $k$-means problem on SBM-based synthetic datasets of sizes $n = 10k, 20k, 50k$, our algorithms reduce the number of strong edge queries by at least $93$ percent compared to Bateni et al. Our algorithm for $k$-means on the MNIST dataset using SVD and $t$-SNE embeddings improves the query complexity by at least $61$ and $99$ percent, respectively.
>
> |   n   |   **% of strong edge queries**  |      |    | **% of strong edge queries** |     |    |  **Improvement (%)**     |      |       |
> |:-----:|:-----------:|:----------:|:----------:|:--------------:|:----------------:|:----------------:|:-------------:|:------:|:-------:|
> |           |   **(Bateni et al.)** |         |         |  **(Ours)** |        |           |        |      |       |
> |       |  $\delta$ = 0.1  |  &nbsp; $\delta$ = 0.2 &nbsp;  |   &nbsp; $\delta$ = 0.3 &nbsp;  |  &nbsp; $\delta$ = 0.1 &nbsp;  |   &nbsp; $\delta$ = 0.2 &nbsp;  |  &nbsp; $\delta$ = 0.3 &nbsp; | $\delta$ = 0.1 | &nbsp; $\delta$ = 0.2 &nbsp;      | &nbsp; $\delta$ = 0.3 &nbsp;  |
> | 10k   |   0.1537  |  0.0614 |  0.8692  | 0.0029  | 0.0030  |  0.0030  | 98 |  94  | 99 |
> | 20k   |   0.0389  |  0.0315 |  0.7340  | 0.0020  | 0.0021  |  0.0024  | 94 | 93 | 99 |
> | 50k   |   0.0268  |  0.0167 |  0.1370  |  0.0009 | 0.0010  |  0.0011  | 96 |93 | 99  |
>
> **Table 1: Performance of $k$-means algorithms on SBM-based synthetic datasets**
>
> |   MNIST   |   **% of strong edge queries**  |      |    | **% of strong edge queries** |     |    |  **Improvement (%)**     |      |       |
> |:-----:|:-----------:|:----------:|:----------:|:--------------:|:--------------:|:--------------:|:-------------:|:------:|:-------:|
> |           |   **(Bateni et al.)** |         |         |  **(Ours)** |        |           |        |      |       |
> |       |  $\delta$ = 0.1  |  &nbsp; $\delta$ = 0.2 &nbsp;  |   &nbsp; $\delta$ = 0.3 &nbsp;  |  &nbsp; $\delta$ = 0.1 &nbsp;  |   &nbsp; $\delta$ = 0.2 &nbsp;  |  &nbsp; $\delta$ = 0.3 &nbsp; | $\delta$ = 0.1 | &nbsp; $\delta$ = 0.2 &nbsp;  | &nbsp; $\delta$ = 0.3 &nbsp;  |
> |  SVD   |  0.0018 | 0.0029 | 0.0019 | 0.0004 | 0.0005 | 0.0007 |  78| 79 | 61 |
> | t-SNE |  0.629 | 0.6263 | 1.3144 | 0.0016 | 0.0023 | 0.0026 | 99 | 99 | 99 |
>
> **Table 2: Performance of $k$-means algorithms on MNIST datasets**
>
> |   n   |   **% of strong edge queries**  |      |    | **% of strong edge queries** |     |    |  **Improvement (%)**     |      |       |
> |:-----:|:-----------:|:----------:|:----------:|:--------------:|:----------------:|:----------------:|:-------------:|:------:|:-------:|
> |           |   **(Bateni et al.)** |         |         |  **(Ours)** |        |           |        |      |       |
> |       |  $\delta$ = 0.1  |  &nbsp; $\delta$ = 0.2 &nbsp;  |   &nbsp; $\delta$ = 0.3 &nbsp;  |  &nbsp; $\delta$ = 0.1 &nbsp;  |   &nbsp; $\delta$ = 0.2 &nbsp;  |  &nbsp; $\delta$ = 0.3 &nbsp; | $\delta$ = 0.1 | &nbsp; $\delta$ = 0.2 &nbsp;      | &nbsp; $\delta$ = 0.3 &nbsp;  |
> | 10k   | 0.2115 | 0.2749 |0.2749  | 0.0199 | 0.0311 | 0.743 | 90 | 88 | 72 |
> | 20k   | 0.158 | 0.0383 | 0.2978 | 0.0120 | 0.0195 |0.0439 | 23 | 49 | 85 |
> | 50k   |  0.0158 |0.0549 | 0.1191 | 0.0048 | 0.0089 | 0.0202 | 69 | 83 |83 |
>
> **Table 3: Performance of $k$-center algorithms on SBM-based synthetic datasets**
>
>
> (**D**): _Possible takeaway from experimental results:_
>
> We validate the empirical performance of our algorithms by comparing them with the experimental results reported in Bateni et al. for synthetic as well as real-world datasets. For experiments on synthetic data, we run our algorithms on a dataset generated using an SBM and report improved performance of our algorithms over those of Bateni et al. One takeaway from these results could be that our algorithms might perform better on synthetic datasets, as reported here for the SBM-based datasets.

---

> > ### Comment · Reviewer_pEt7 · 2025-08-03
> >
> > Thanks for the responses. I still feel that the model is nice mathematically, but the paper lacks a solid justification as the motivations coming from crowd sourcing or deep networks fits the model only partially (whether different pairs are independent, or querying the same pair multiple times). I think the direction is important and thus confirm my score.

---

### Official Review · Reviewer_vyMs · 2025-06-30

**Clarity:** 3
**Significance:** 3
**Originality:** 2
**Rating:** 3
**Confidence:** 3

**Summary:**

The authors develop approximation algorithms for clustering problem in the strong and weak oracle model, where an oracle provides exact or noisy distances between the input points, respectively. Their theoretical results are complemented with an experimental evaluation on real-world datasets.

**Questions:**

- the authors claim that one of the main contributions of their work is to remove the assumption that the aspect-ratio be polynomial which is needed in [6]. However, in appendix A of [6] the authors claim that this can be easily removed. Could the authors please clarify?

**Ethical Concerns:**

["NO or VERY MINOR ethics concerns only"]

**Final Justification:**

I still feel the results are interesting but they provide only marginal improvement over the state of the art.

**Limitations:**

yes

**Paper Formatting Concerns:**

No formatting issues

**Quality:**

3

**Strengths And Weaknesses:**

Strenghts:
- the problems studied in the paper are interesting and relevant to the NeurIPS community
- some of the theoretical results are novel and interesting
- an experimental evaluation on real-world datasets is included in the paper

Weaknesses:
- some of the results show only a marginal improvement over previous ones (in particular they remove some factors which are polynomial in k and logn while introducing a dependency on 1/epsilon^6 )
- some of their claims appear not to be accurate (see below), as a result it is difficult to appreciate their novel technical contributions overall

Some other comments:
- the fact that k-means++ can be adapted to the weak-oracle model is interesting, however, to be fair there are well-known algorithms based on coreset for this problem
- the lower bound on the weak-oracle model is novel to best of my knowledge and perhaps it should be stressed more in the paper

---

> ### Author Rebuttal · Authors · 2025-07-28
>
> We thank the reviewer for making insightful comments about our work. We tried to address the concerns raised by the reviewer below. We will be happy to provide more details if the reviewer asks for them.
>
> (**A**): _``the authors claim that one of the main contributions ...aspect-ratio be polynomial which is needed in [6]. ... Could the authors please clarify?'':_
>
> We highlight the fact that the algorithms of Bateni et al. guess the value of the optimal solution. Even though this is a common assumption and routinely used in the design of approximation algorithms, one incurs some overhead because of this. As noted by the reviewer, Bateni et al. in Appendix A argue how to overcome this assumption using a guess-and-verify approach. They incur a multiplicative factor (a function of the aspect ratio, $\log{\frac{\log{\Delta}}{\varepsilon}}$) in their query complexity. Our comment was made in the context that our algorithms would not have this dependency on the aspect ratio.
>
> We consider this to be a minor improvement of our algorithms over that of Bateni et al. and not the main contribution. If the reviewer feels that removing this aspect will allow the reader to focus on the other main contributions, we will be happy to do that.
>
>
> (**B**): _Query regarding the extent of improvement of theoretical results:_
>
> Our algorithm for $k$-means uses $O(\frac{k^2 \log^2 n}{\epsilon^6})$ strong-oracle queries and $O(\frac{nk \log n}{\epsilon^3})$ weak-oracle queries and provides a $40(1+\epsilon)$-approximation using $O(k\frac{\log n}{\epsilon^3})$ centers. This is in contrast to Bateni et al. who use $O(k^2 \log^4 n \log^2(\frac{\log n}{\epsilon}))$ strong-oracle queries and $O(nk \log^2 n \log(\frac{\log n}{\epsilon}))$ weak-oracle queries and provides a $O(1)$-approximation using $O(\log^2 n \log(\frac{\log n}{\epsilon}))$ centers. Overall, our algorithm provides an improved approximation factor while having a better dependency on $\log n$ on the query complexity, but a worse dependency on $\epsilon$. Our algorithm might be favoured to be used when $\epsilon$ is not too small.
>
> (**C**): _``...the fact that k-means++ can be adapted to the weak-oracle model is interesting, ... well-known algorithms based on coreset for this problem'':}:_
>
> $k$-means++ is one of the most widely used algorithms for $k$-means in practice. There have been many attempts [7,8,11,16] that study the robustness of the $k$-means++ algorithm in the presence of outliers or when the sampling distribution is noisy. In this work, we show that $k$-means++ can be adapted to work in the weak-strong oracle model. This adds to the justification of using $k$-means++ in practice. To the best of our knowledge, the only coreset-based solution for $k$-means is given by Bateni et al. in the weak-strong oracle model.
>
> (**D**): _``... the lower bound on the weak-oracle model is novel to best of my knowledge and perhaps it should be stressed more in the paper'':_
>
> We are glad that the reviewer liked the result; we will emphasise it more in the paper.

---

> > ### Comment · Reviewer_vyMs · 2025-08-05
> >
> > Thanks for your response and for clarifying those points. I will stand by my score.

---

### Official Review · Reviewer_aeyT · 2025-07-02

**Clarity:** 3
**Significance:** 3
**Originality:** 3
**Rating:** 5
**Confidence:** 4

**Summary:**

This paper provides new improved algorithms for solving $k$-means and $k$-center problems in the *weak-strong oracle model* recently introduced in a COLT 2024 paper by Bateni et al. In this weak-strong oracle model, the point embeddings in $\mathbb{R}^d$ are not available directly but instead for any $2$ points $x,y$, there is an option to either do a *strong-oracle* query $SO(x,y)$ which returns the exact distance between the points $d(x,y)$, or do a *weak-oracle* query $WO(x,y)$ which returns the exact distance $d(x,y)$ with probability $1-\delta$ or returns some arbitrary value with probability $\delta$ for some fixed $\delta \in (0,1/2)$. This is an interesting model, especially for modern times in the era of LLMs, where it might be much cheaper to get noisy information from very cheap (or free) queries from less expensive models (like GPT-4o) while it might be extremely expensive for any single query to more high-quality models (like o1 pro). The submission provides algorithms with both, better theoretical approximation factors, and also better empirical performance than the corresponding algorithms in Bateni et al.

**Questions:**

Regarding the experiments, one qualm I have is that since the entire current paper never discussed point queries in the theoretical portion, it seems unfair to impose them on the readers in the experimental section. I did not read Bateni et al. earlier so it was unnecessary to have the exact comparison using their reported values. You could have run your own experiments using the algorithms of Bateni et al. on the datasets to compute the number of strong-oracle queries which you have been discussing about through your entire paper.

Also, regarding the choice of constant probability bi-criteria bounds in the footsteps of Aggarwal et al 2009: There have been follow up works on getting expectation bounds for bi-criteria approximation by Wei in NeurIPS 2016 [1] and Makarychev et al in NeurIPS 2020 [2]. In general, I am sure you are aware of the several reasons why proving bounds on the expectation are the defacto choice in the approximation algorithms literature over proving constant probability bounds. Please go over these papers and at least make a comment at the end of your paper that your results could be improved to expectation bounds following in the footsteps of these papers.

1) "A Constant-Factor Bi-Criteria Approximation Guarantee for k-means++" by Wei, NeurIPS 2016
2) "Improved Guarantees for k-means++ and k-means++ Parallel" by Makarychev et al, NeurIPS 2020.

**Ethical Concerns:**

["NO or VERY MINOR ethics concerns only"]

**Final Justification:**

The authors have sufficiently addressed my concerns regarding the comparison with prior work and so I would like to increase my score.

**Limitations:**

Yes

**Quality:**

3

**Strengths And Weaknesses:**

Quality: Overall the submission is technically sound. The theoretical claims are well-supported with proofs provided in the Appendix. I have gone over the algorithm descriptions along with the intuitions for the proofs given in the main paper and they seem to be sound. The experimental results are also well-documented and seem to be okay (but I have a qualm which I will mention in the questions section of the review).

Clarity: Overall clearly written paper. But I urge the authors to make an effort to chop up the huge wall-of-text paragraphs that are flowing in the paper in several sections like subsection 1.3 and section 3. I understand you have the 9-page constraint to which you have to stick, but in the current form note that potential readers (especially new grad students who might be unfamiliar with a lot of the arguments you are making) might find these huge walls-of-text very daunting to read.

Significance: Yes, the results of the paper are significant to the community in my opinion. This weak-strong oracle model which this paper is studying seems to be interesting and potentially useful so I believe there will be interest from other researchers.

Originality: The weak-strong oracle model is a very new model. Therefore it is unsurprising that not much work has been done on this model yet. The results in this paper seem to definitely be decent improvements over the original paper that proposed this model. Now whether these results are very original is a different question altogether. Majority of the arguments used in the paper seem to follow from the standard literature on clustering. One of the key contributions of this paper is in analyzing k-means++ in this model, which doesn't seem to be the approach used by the earlier paper...

---

> ### Author Rebuttal · Authors · 2025-07-28
>
> We thank the reviewer for the insightful questions. Please find below our replies to the queries.
>
> (**A**): _Discussion on the use of point queries and edge (pair) queries:_
>
> In this paper, we chose to work with only weak and strong edge queries (or pair queries) in the weak-strong oracle model. We believe that assuming access to point queries makes a stronger assumption on the model. To recall, a point query gives the complete embedding of a point, whereas an edge query only gives the distance between the two points. For any set of $t$ points, one can make either $t$ point queries or ${t\choose 2}$ edge queries to obtain the complete distance information between the points. We highlight that it is possible to rewrite our algorithms to use point queries instead of edge queries, and these algorithms would use fewer point queries compared to the algorithms of Bateni et al.
>
> (**B**): _Experimental results using edge queries:_
>
> Our algorithms for both $k$-means and $k$-center sample points of the input, and use strong-edge queries to obtain exact distances between the sampled points. Had we used point queries in our algorithms, the same guarantees could be achieved by making a point query for each sampled point. We make strong-edge queries to obtain distances between any two sampled points, whereas Bateni et al. makes a point query for each sampled point. Let $t_1$ and $t_2$ denote the number of points sampled by our algorithm and Bateni et al., respectively. Then, our algorithm makes ${t_1\choose 2}$ strong-edge queries, whereas Bateni et al. make $t_2$ point queries.
>
> Since some reviewers became confused with our use of strong edge queries in the descriptions of algorithms, and the use of point queries to state experimental results, we decided to report our experimental results using the number of strong edge queries used by the algorithms. Since Bateni et al. only report the number of point queries, we calculate the equivalent number of strong edge queries and compare them with our results.
>
>
> Our algorithms use substantially fewer strong edge queries compared to algorithms of Bateni et al. We report the percentage of strong edge queries used by our algorithms and those of Bateni et al. for both $k$-means and $k$-center problems for both synthetic and real-world datasets, and for different choices of the failure probability $\delta$. For the $k$-means problem on SBM-based synthetic datasets of sizes $n = 10k, 20k, 50k$, our algorithms reduce the number of strong edge queries by at least $93$ percent compared to Bateni et al. Our algorithm for $k$-means on the MNIST dataset using SVD and $t$-SNE embeddings improves the query complexity by at least $61$ and $99$ percent, respectively.
>
> |   n   |   **% of strong edge queries**  |      |    | **% of strong edge queries** |     |    |  **Improvement (%)**     |      |       |
> |:-----:|:-----------:|:----------:|:----------:|:--------------:|:----------------:|:----------------:|:-------------:|:------:|:-------:|
> |           |   **(Bateni et al.)** |         |         |  **(Ours)** |        |           |        |      |       |
> |       |  $\delta$ = 0.1  |  &nbsp; $\delta$ = 0.2 &nbsp;  |   &nbsp; $\delta$ = 0.3 &nbsp;  |  &nbsp; $\delta$ = 0.1 &nbsp;  |   &nbsp; $\delta$ = 0.2 &nbsp;  |  &nbsp; $\delta$ = 0.3 &nbsp; | $\delta$ = 0.1 | &nbsp; $\delta$ = 0.2 &nbsp;      | &nbsp; $\delta$ = 0.3 &nbsp;  |
> | 10k   |   0.1537  |  0.0614 |  0.8692  | 0.0029  | 0.0030  |  0.0030  | 98 |  94  | 99 |
> | 20k   |   0.0389  |  0.0315 |  0.7340  | 0.0020  | 0.0021  |  0.0024  | 94 | 93 | 99 |
> | 50k   |   0.0268  |  0.0167 |  0.1370  |  0.0009 | 0.0010  |  0.0011  | 96 |93 | 99  |
>
> **Table 1: Performance of $k$-means algorithms on SBM-based synthetic datasets**
>
> |   MNIST   |   **% of strong edge queries**  |      |    | **% of strong edge queries** |     |    |  **Improvement (%)**     |      |       |
> |:-----:|:-----------:|:----------:|:----------:|:--------------:|:----------------:|:----------------:|:-------------:|:------:|:-------:|
> |           |   **(Bateni et al.)** |         |         |  **(Ours)** |        |           |        |      |       |
> |       |  $\delta$ = 0.1  |  &nbsp; $\delta$ = 0.2 &nbsp;  |   &nbsp; $\delta$ = 0.3 &nbsp;  |  &nbsp; $\delta$ = 0.1 &nbsp;  |   &nbsp; $\delta$ = 0.2 &nbsp;  |  &nbsp; $\delta$ = 0.3 &nbsp; | $\delta$ = 0.1 | &nbsp; $\delta$ = 0.2 &nbsp;  | &nbsp; $\delta$ = 0.3 &nbsp;  |
> |  SVD   |  0.0018 | 0.0029 | 0.0019 | 0.0004 | 0.0005 | 0.0007 |  78| 79 | 61 |
> | t-SNE |  0.629 | 0.6263 | 1.3144 | 0.0016 | 0.0023 | 0.0026 | 99 | 99 | 99 |
>
> **Table 2: Performance of $k$-means algorithms on MNIST datasets**
>
> |   n   |   **% of strong edge queries**  |      |    | **% of strong edge queries** |     |    |  **Improvement (%)**     |      |       |
> |:-----:|:-----------:|:----------:|:----------:|:--------------:|:----------------:|:----------------:|:-------------:|:------:|:-------:|
> |           |   **(Bateni et al.)** |         |         |  **(Ours)** |        |           |        |      |       |
> |       |  $\delta$ = 0.1  |  &nbsp; $\delta$ = 0.2 &nbsp;  |   &nbsp; $\delta$ = 0.3 &nbsp;  |  &nbsp; $\delta$ = 0.1 &nbsp;  |   &nbsp; $\delta$ = 0.2 &nbsp;  |  &nbsp; $\delta$ = 0.3 &nbsp; | $\delta$ = 0.1 | &nbsp; $\delta$ = 0.2 &nbsp;      | &nbsp; $\delta$ = 0.3 &nbsp;  |
> | 10k   | 0.2115 | 0.2749 |0.2749  | 0.0199 | 0.0311 | 0.743 | 90 | 88 | 72 |
> | 20k   | 0.158 | 0.0383 | 0.2978 | 0.0120 | 0.0195 |0.0439 | 23 | 49 | 85 |
> | 50k   |  0.0158 |0.0549 | 0.1191 | 0.0048 | 0.0089 | 0.0202 | 69 | 83 |83 |
>
> **Table 3: Performance of $k$-center algorithms on SBM-based synthetic datasets**
>
> (**C**): _Whether our results can be extended to give constant factor bi-criteria approximation in expectation for $k$-means:_
>
> Thanks for this suggestion. Yes, we believe the expectation bounds of Wei and Makarychev et al should also apply. However, the main advantage of analysing expectation instead of high-probability bounds was that the former allows one to reduce the number of centers sampled. Note that in the current context, we pay a multiplicative factor of (log n) on the number of samples just to get good distance estimates. So, analysing the expectation is less attractive in this setting. We will definitely add these points to the discussions.
>
> (**D**): _Regarding wall-of-text paragraphs in Section 1.3 and Section 3._
>
> Thanks for the suggestion. We will try to format it in a manner that makes it easier to read.

---

> > ### Comment · Reviewer_aeyT · 2025-08-08
> >
> > The authors have sufficiently addressed my concerns regarding the comparison with prior work and so I would like to increase my score. I expect them to include these results and a more complete discussion of the comparison in their revision (at least in the supplementary, if there isn't enough space in the main paper). I have gone over the submitted supplementary now and it seems like some huge wall-of-text paragraphs also exist in the supplementary. I suggest the authors to break some of them into multiple shorter paragraphs for better readability.

---

### Decision · Program_Chairs · 2025-09-17

**Decision:**

Reject

**Comment:**

The paper studies clustering in the model with two distance oracles: one always correct and one can be arbitrarily wrong with a certain probability. The paper gives new algorithms for k-means and k-centers in this model with a different tradeoff compared with previous work in the number of queries. The new number of queries for k-means saves several factors of log(n) compared with previous work. The new number of queries for k-centers saves two log(n) factors but has an extra factor of k compared with previous work.

The reviewers appreciate this clean model, though they also think the model does not align well with practical applications where oracle answers come from a deep model. Another concern is that the experiments are limited to a synthetic dataset. The comparison is limited to only the number of queries but not the solution accuracy.

Overall, the majority of the reviewer are slightly positive and one reviewer is slightly negative. Given that there are no reviewers strongly supporting the paper and there are some significant downside, the paper unfortunately cannot compete with other submissions with stronger supports.